# Potency Evaluations of Recombinant Botulinum Neurotoxin A1 Mutants Designed to Reduce Toxicity

**DOI:** 10.3390/ijms25168955

**Published:** 2024-08-17

**Authors:** Polrit Viravathana, William H. Tepp, Marite Bradshaw, Amanda Przedpelski, Joseph T. Barbieri, Sabine Pellett

**Affiliations:** 1Department of Bacteriology, University of Wisconsin-Madison, Madison, WI 53706, USA; 2Department of Microbiology and Immunology, Medical College of Wisconsin, Milwaukee, WI 53226, USA

**Keywords:** botulinum neurotoxin, BoNT, expression system, endogenous expression, atoxic, recombinant

## Abstract

Recombinant mutant holotoxin BoNTs (rBoNTs) are being evaluated as possible vaccines against botulism. Previously, several rBoNTs containing 2–3 amino acid mutations in the light chain (LC) showed significant decreases in toxicity (2.5-million-fold–12.5-million-fold) versus wild-type BoNT/A1, leading to their current exclusion from the Federal Select Agent list. In this study, we added four additional mutations in the receptor-binding domain, translocation domain, and enzymatic cleft to further decrease toxicity, creating 7M rBoNT/A1. Due to poor expression in *E. coli*, 7M rBoNT/A1 was produced in an endogenous *C. botulinum* expression system. This protein had higher residual toxicity (LD50: 280 ng/mouse) than previously reported for the catalytically inactive rBoNT/A1 containing only three of the mutations (>10 µg/mouse). To investigate this discrepancy, several additional rBoNT/A1 constructs containing individual sets of amino acid substitutions from 7M rBoNT/A1 and related mutations were also endogenously produced. Similarly to endogenously produced 7M rBoNT/A1, all of the endogenously produced mutants had ~100–1000-fold greater toxicity than what was reported for their original heterologous host counterparts. A combination of mutations in multiple functional domains resulted in a greater but not multiplicative reduction in toxicity. This report demonstrates the impact of production systems on residual toxicity of genetically inactivated rBoNTs.

## 1. Introduction

Neurotoxins produced by anaerobic, spore-forming bacteria of the genus *Clostridium* include the most potent biological toxins known on the planet, botulinum neurotoxins (BoNTs) and tetanus toxin (TeNT). BoNTs are organized into seven serotypes (BoNT/A-BoNT/G) and at least 40 subtypes [1,2,3]. These 150 kDa multi-domain enzymes comprise an N-terminal, zinc-dependent protease light chain (LC, ~50 kDa), and dual-function heavy chain (HC; ~100 kDa) [2,4]. The heavy chain is composed of an N-terminal (HN) translocation domain, which aids LC entry into the neuronal cell cytosol, while the C-terminal domain of the heavy chain (HC) is responsible for specific neuronal cell surface receptor recognition [5]. The LC and HC are connected by a linker region in the 150 kDa protoxin that is cleaved by endogenous or exogenous proteases to convert the toxin to the disulfide-linked, more active dichain form [6,7,8].

BoNTs specifically enter peripheral neurons, primarily motor-neurons, where they block the release of acetylcholine by the synaptic vesicle [9,10]. The HC binding domain interacts with gangliosides and synaptic vesicle protein receptors of the neuronal presynaptic membrane, allowing the BoNT to enter the neuron via an endocytic vesicle [11,12]. Acidification of the endocytosed vesicle or endosome leads to a conformational change in the endocytosed toxin and HC translocation domain, forming a channel in the vesicle envelope, permitting the LC to enter the neuronal cytosol, where it enzymatically cleaves specific soluble *N*-ethylmaleimide-sensitive factor-activating protein receptor (SNARE) proteins, depending on serotype [13,14,15,16]. The resulting cleavage of a SNARE protein prevents formation of a functional SNARE complex and thereby renders the acetylcholine-filled synaptic vesicles incapable of fusing with the cell membrane to release their cargo. The absence of acetylcholine signaling causes muscles to be unable to contract resulting in the flaccid paralysis typical of botulism [17].

BoNTs are a Category A Tier 1 Select Agent due to their potency, lack of a cure, and potential misuse as a bioterrorism weapon [18,19,20]. BoNT serotypes A, B, E, and F cause naturally occurring botulism in humans, where BoNT/A causes the most severe and longest lasting disease [17,21]. As of 2011, there is no available US vaccine approved for human use. Prior to 2011, a pentavalent BoNT toxoid preparation against botulism serotypes A-E, produced by the Michigan Department of Health (MDPH), was available as an investigational drug, but it was discontinued in 2011 due to decreasing efficacy and increasing side effects [22,23,24,25]. Therefore, a new vaccine against botulism would be beneficial to individuals at risk of acquiring botulism.

Recent developments of a new botulism vaccine have focused on either a recombinant non-toxic subunit or a catalytically inactive full-length BoNT holotoxin (rBoNT). While the full-length rBoNT offers the benefit of presenting the greatest number of epitopes, thereby facilitating optimal immunity induction, it also poses the significant challenge of potential harmful residual toxicity. Traditionally, this is overcome by inactivating and denaturing the protein toxin with a 2-week formalin treatment, but that treatment is associated with undefined changes in the protein toxins, potentially reducing immunogenicity, and residual formalin in the vaccine can result in unpleasant side effects. Thus, recent efforts have focused on genetically inactivating BoNTs through targeted amino acid substitutions in key functional domains as a novel vaccine [25]. Recombinant protein expression can be carried out via a variety of prokaryotic and eukaryotic expression systems. Several full-length recombinant BoNT/A1 mutant toxins have been successfully produced in *E. coli*, *C. botulinum*, *Pichia pastoris*, and Sf9 insect cell expression systems [26,27,28,29]. Some of these were shown to be promising vaccine candidates against BoNT/A1, with excellent immunogenicity and from 2.5-million-fold to 10-million-fold reduced residual toxicity in mice [25,28,29,30,31,32,33].

Multiple studies have shown that the specific introduction of targeted mutations in functional domains of BoNT/A1 reduced the respective toxin related function in vitro or in vivo. Several studies investigated BoNTs with 2–3 amino acid residue mutations in the LC that are critical for zinc coordination. rBoNT/A1^R363A,Y365F^ produced in *E. coli* had no detectable residual toxicity in mice at 1 mg and showed promise as a novel vaccine [28]. Another study produced rBoNT/A1^E224A,E262A^ in *E. coli* (termed DR BoNT/A) and reported a >10-million-fold reduction in toxicity in mice [33,34,35,36]. A similar mutant produced in Sf9 insect cell, rBoNT/A1^E224A,Y366A^ (termed BoNT/A ad), did not result in SNAP-25 cleavage in rat spinal cord cells at doses of ~300 nM in initial studies, but follow-up studies reported SNAP-25 cleavage in rat hippocampal neurons at >100 pM [26,37]. In the intraperitoneal mouse LD50 assay, BoNT/A ad resulted in an LD50 of ~1 µg for the dichain form of the toxin and ~9 µg for the single chain, a 100,000-fold and 1-million-fold decrease compared to wild-type BoNT/A1, respectively [26,27]. Finally, full-length rBoNT/A1^E224A,R363A,Y366F^ produced in *E. coli* had a greater than 2-million-fold decrease in potency versus wild-type BoNT/A1 when compared in in vivo testing in mice [30]. Webb and associates produced BoNT/A-F with two histidines and a glutamate in the LC mutated to prevent zinc coordination (H223A, E224A, and H227A), designated ciBoNT/A–F. These recombinant proteins were produced in *Pichia pastoris* (yeast) and resulted in a greater than 2-million-fold decrease in toxicity, with an about 20-million-fold decrease for ciBoNT/A. The ciBoNTs also provided effective protection against BoNT/A challenge in mice [29]. The introduction of a single-point mutation in the ganglioside binding domain (W1266L) of a rBoNT/A1 HC fragment produced in *E. coli* neutralized its binding capacity to gangliosides in vitro by about 100-fold and toxicity, as measured by mouse phrenic nerve assay, by ~140-fold [38]. This was further confirmed in neuronal cell entry studies with an *E. coli*-produced BoNT/A1 HC_C_ fragment in which the key tryptophan residue was changed to an alanine (W1266A), resulting in a lack of entry into rat cortical neurons [39]. In a small pilot vaccine study the mutated rBoNT/A1 HC_C_^W1266A^ resulted in similar or even slightly greater protection compared to wild-type HC_C_ [39]. In another strategy, a double mutation (L175A and D370A) was introduced into the enzymatic cleft of the BoNT/A light chain, which decreased its hydrolytic activity in an in vitro SNAP-25 cleavage assay by >40,000-fold [40]. Several of these recombinant BoNT/A1 holotoxins are currently excluded from the US Federal Select Agent List on the basis of the reported strongly reduced toxicity [19].

In this study, we examined the impact of these and additional amino acid changes in BoNT/A1 functional domains of full-length tag-less rBoNT/A1 produced in an endogenous *C. botulinum* expression host using two biologic assays, the mouse bioassay and a cell-based assay. Furthermore, we evaluated a combination of amino acid changes in multiple functional domains of BoNT/A1 to determine whether multiple layers of mutations could further reduce residual toxicity.

Our data revealed that all the endogenously produced tag-less mutant BoNT/A1 toxins exhibited 100–1000-fold higher toxicity compared to their counterparts produced in a heterologous host, as reported in previous studies. Furthermore, all recombinant BoNT/A mutants were capable of entering cultured neurons and cleaving SNAP-25 to varying degrees. These results demonstrate that while targeted mutations of specific amino acids in functional domains of BoNT/A1 can decrease toxicity, the presence of protein tags or other modifications and the production in heterologous hosts can also significantly contribute to decreases in toxicity. Furthermore, this study reiterates the importance of testing residual toxicity in biological systems.

## 2. Results

### 2.1. Catalytically Inactive Mutated rBoNT/A1 Produced in C. botulinum Cleaves SNAP-25 and Causes In Vivo Toxicity at 280 ng

Previous studies indicated that rBoNT/A1^E224A,R363A,Y366F^ had low residual toxicity (reduced >1-million-fold) and was an excellent vaccine candidate [30]. An additional mutation in the ganglioside binding pocket did not diminish vaccine efficacy [30]. In order to add additional layers of safety, 7M rBoNT/A1 was designed with mutations in the translocation domain (K759A) and the ganglioside binding domain (W1266A) of the heavy chain, as well as mutations targeting the enzymatic capacity (L175A and D370A) and zinc-coordination capability (E224Q, R363A, and Y366F) of the light chain (Figure 1). Moreover, 7M BoNT/A1 and wild-type BoNT/A1 subjected to superposition/alignment in PyMol [41] yielded a RMSD (root mean square deviation) of 0.311 Å between the two proteins, indicating that the overall structure of 7M BoNT/A1 remained relatively unchanged compared to the wt BoNT/A1 structure (Figure 1).

Attempts to produce 7M rBoNT/A1 in *E. coli* at sufficient quantities for toxicity and vaccination studies were unsuccessful due to the protein being insoluble. However, 7M rBoNT/A1 was successfully produced in an endogenous clostridial expression system, which also produces all BoNT/A complexing proteins but no wild-type toxin [44]. Since protocols for purification of BoNT/A from *C. botulinum* by a series of precipitations and chromatography steps are well established [45], no protein tags or other modifications were added to the endogenously produced 7M rBoNT/A1. SDS-PAGE analysis showed a single 150 kDa band under non-reducing conditions, and 100 kDa and 50 kDa bands in a reduced sample (Figure 2A), indicating that endogenously produced 7M rBoNT/A1 was converted to the dichain form by endogenous proteases. An in vivo toxicity analysis by mouse assays resulted in an LD50 (mouse lethal dose 50) of the 7M rBoNT/A1 at 280 ng (Figure 2B).

This is in contrast to previously reported LD50s for rBoNT/A1 mutants containing only some of the amino acid alterations included in 7M BoNT/A1, such as *E. coli*-produced rBoNT/A^E224Q,R363A,Y366F^ and yeast produced ciBoNT/A, at >10 and >50 µg, respectively [29,30]. The ability of 7M rBoNT/A1 to enter neuronal cells and cleave intracellular SNAP-25 was examined via the exposure of highly sensitive primary rat spinal cord cells to serial dilutions of 7M rBoNT/A1 in parallel with exposure to wt BoNT/A1 (Figure 2D). The 7M BoNT/A1 resulted in clear dose-dependent SNAP-25 cleavage at concentrations above ~7.5 ng and a maximum cleavage reached at ~60% of SNAP-25 (Figure 2C).

The ~70,000-fold reduction in toxin potency of 7M rBoNT/A1 compared to wt BoNT/A1, as well as its ability to enter neurons and cleave SNAP-25, was surprising and inconsistent with previous reports of mutated rBoNT/A1, where a >1-million-fold reduction in potency was reported and no or little SNAP-25 cleavage was shown [26,28]. Since this study used an endogenous expression host and tag-less BoNT/A1, we felt it was prudent to examine whether previously reported data on reduced toxicity were due to the reported amino acid alterations or to other factors. Thus, the impact on toxicity of each set of mutation(s) that comprised 7M rBoNT/A1s, as well as some additional related mutants, was assessed using rBoNTs synthesized via the endogenous *Clostridium botulinum* BoNT/A1 expression system [44].

### 2.2. Mutations of K759 in the BoNT/A1 Translocations Domain Does Not Reduce Toxicity

A key lysine residue (K768) in the translocation domain of the related clostridial tetanus neurotoxin has previously been shown to be essential for translocation [42]. The equivalent amino acid alteration introduced into BoNT/A1 and expressed in *C. botulinum* resulted in a dichain rBoNT/A1^K759A^ (Figure 3A) with a mouse LD50 of 6 pg/mouse, similar to the 4 pg/mouse observed for BoNT/A1 wt (Figure 3B). Likewise, no difference in potency was observed in a comparative cell-based assay using primary rat spinal cords between BoNT/A1 wt and rBoNT/A1^K759A^ (Figure 3C,D).

These data indicate that while the K768 residue in tetanus toxin is essential for proper translocation, the homologous residue in BoNT/A1 does not affect toxin cell entry.

### 2.3. Mutations of Key Amino Acids in the BoNT/A1 Ganglioside Binding Domain and SV2 Receptor Binding Domain Reduced Toxicity by ~25-Fold to ~90-Fold

Altering of the W1266 residue of the ganglioside-binding domain of the BoNT/A1 HC resulted in an inability of the recombinant receptor-binding domain to enter neurons in vitro and an ~140-fold decrease in potency versus wild-type BoNT/A1 in the in vitro mouse hemidiaphragm assay [38,39]. A single amino acid substitution (G1292R) in the SV2 binding site of BoNT/A1 was reported to reduce potency 300-fold in the hemidiaphragm assay [46]. All of these mutants were produced in *E. coli*. *Clostridium*-produced rBoNT/A1^W1266A^ (Figure 4A) resulted in a mouse LD50 of 350 pg/mouse, indicating an ~90-fold decrease in toxin potency compared to wt BoNT/A1, and *clostridium*-produced rBoNT/A1^G1292R^ resulted in a mouse LD50 of 100 pg/mouse, a 25-fold decrease compared to wt BoNT/A1 (Figure 4B). Both rBoNT/A1^G1292R^ and rBoNT/A1^W1266A^ resulted in dose-dependent SNAP-25 cleavage in primary rat spinal cord neurons with EC50s of ~800 pg, compared to an EC50 of ~7 pg for wt BoNT/A1 and reaching 100% SNAP-25 cleavage (Figure 4C,D), indicating the mutated toxin’s ability to enter neurons, albeit at lower efficiency.

These data indicate that the W1266A and G1292R mutations impair but do not eliminate neuronal cell entry and SNAP-25 cleavage by BoNT/A.

### 2.4. Mutations Preventing Zinc Coordination in the BoNT/A1 Light Chain Reduced Toxin Potency ~12,500-Fold and Mutation of Key Residues in the Synaptic Cleft Reduced Potency ~2000-Fold

The 50 kDa light chain of BoNT/A1 is a zinc-dependent metalloprotease, and removal of zinc has been shown to eliminate enzymatic activity [47,48]. Similarly, the mutation of key amino acid residues involved in zinc coordination (E224Q, R363A, Y366F or H223A, E224Q, and H227A) has been shown to result in a >1-million-fold reduction of potency in mice [29,30]. In another study, a leucine (175) and aspartic acid (370) located in the synaptic cleft of the LC were found to be essential for in vitro enzymatic activity of the BoNT/A1 LC, and the mutation of both residues to alanine reduced SNAP-25 cleavage by >40,000-fold in an in vitro enzymatic assay [40]. In all of these studies, the proteins had been produced in *E. coli* or *Pichia pastoris*. Production of rBoNT/A1^L175A,D370A^, rBoNT/A1^E224Q,R363A,Y366F^, and ciBoNT/A1^H223A,E224Q,H227A^ in our *C. botulinum* expression system yielded dichain proteins (Figure 5A) with mouse LD50s of 7 ng/mouse (rBoNT/A1^L175A,D370A^) and 50 ng/mouse (rBoNT/A1^E224Q,R363A,Y366F^ and ciBoNT/A1^H223A,E224Q,H227A^) (Figure 5B). All three mutant proteins were able to enter primary rat spinal cord neurons and cleave SNAP-25; however, only rBoNT/A1^L175A,D370A^ reached 100% SNAP-25 cleavage, whereas maximum SNAP-25 cleavage leveled off at about 60–70% for rBoNT/A1^E224Q,R363A,Y366F^ and ciBoNT/A1^H223A,E224Q,H227A^, similar to 7M rBoNT/A1 (Figure 5C,D). The rBoNT/A1^L175A,D370A^ had an EC50 of approximately 15 ng/well in the rat spinal cord cell assay, compared to 7 pg/well for BoNT/A1 wt (Figure 5C).

As with 7M rBoNT/A1, the EC50 of rBoNT/A1^E224Q,R363A,Y366F^, and ciBoNT/A1^H223A,E224Q,H227A^ could not be calculated due to failure to reach 100% SNAP-25 cleavage, resulting in an incomplete dose–response curve compared to wt BoNT/A1, but the 50% SNAP-25 cleavage point can be estimated to be around 100 ng/well (Figure 5C,D), about ~14,000-fold greater than the EC50 of wt BoNT/A1 in the same assay. These data indicate that the mutations preventing zinc coordination dramatically reduce but do not entirely eliminate enzymatic activity of the BoNT/A1 LC, and that the L175A/D370A double mutation in the enzymatic cleft reduces enzymatic activity by about 2000-fold.

## 3. Discussion

This study investigated the effect of single and combined targeted amino acid substitutions in functional domains of endogenously produced rBoNT/A1 on toxicity. Several of these amino acid substitutions had previously been examined using rBoNT/A1 produced in heterologous hosts [30,38,39,40,42]. In contrast, this study utilized an atoxic *C. botulinum* expression system to produce mutated rBoNT/A1 in an endogenous host [44]. All endogenously produced mutated rBoNT/A1 proteins analyzed in this study had greater residual toxicity than was previously reported for their counterparts produced in heterologous hosts (Table 1).

In addition, even though the mutations were designed based on in vitro data indicating a very strong reduction in or elimination of the respective toxin function, all mutant rBoNTs retained residual toxicity in the range of approximately 25–70,000-fold compared to native, wild-type BoNT/A1 produced in the same manner.

The mechanism of how BoNT/A1 enters the neuron and disrupts acetylcholine based cellular communication is a multi-step process. The benefits of cell-based assays include the small amount of the toxin required to perform the test and the high sensitivity and specificity. Also, each step in the BoNT/A1 intoxication process is required to occur in the cell-based assay, including interaction of the toxin with the neuronal cell surface receptors, endocytosis, translocation of the LC into the cell cytosol, and SNARE cleavage [46]. In cell-based assays, all mutant rBoNT/A1 toxins tested in this study maintained the ability to enter neuronal cells and cleave SNAP-25 proteins, albeit at a lower potency when compared to wild-type BoNT/A1 (Figure 2, Figure 3, Figure 4 and Figure 5), indicating reduction but not elimination of the steps required in the cellular intoxication pathway. Most notably, the *C. botulinum* produced variants ciBoNT/A1 and rBoNT/A1^E224Q,R363A,Y366F^, both of which are currently exempt from Select Agent regulations based on previously reported >10-million-fold reduced toxicities using rBoNTs produced in heterologous hosts [29,30], had an estimated ~10,000-fold reduced potency in primary rat spinal cord cells (Figure 5). Similar results were obtained with the in vivo mouse bioassay (Figure 5 and Table 1). A similar trend, although less pronounced, was also observed with additional mutant rBoNT/A1s, including rBoNT/A1^W1266A^ and rBoNT/A1^G1292R^, where the *C. botulinum* produced proteins had an ~2-fold and 10-fold lower reduction in toxin potency than their heterologously produced counterparts, which was estimated by the mouse phrenic nerve assay [46]. The discrepancy in residual toxicity of mutant rBoNT/A1 produced in a *C. botulinum* host versus a heterologous host could be due to several reasons.

Botulinum neurotoxins produced by proteolytic *C. botulinum*, such as BoNT/A1, are known to undergo post-translational proteolytic processing to convert the 150 kDa protoxin to the 50 kDa LC and 100 kDa HC dichain, which has increased activity compared to the unprocessed protoxin [51]. While this process does not occur in heterologous hosts, it has been taken into consideration during toxicity analyses, where the rBoNT/A1 proteins are converted to the more active dichains in vitro by mild trypsin digestion before toxicity testing [8,52]. Interestingly, a wt rBoNT/A1 with similar toxicity as native BoNT/A1 has been successfully produced in *E. coli* [49,50]. Unlike many of the mutant heterologously produced rBoNT/A1s, which typically contain C- or N-terminal protein tags for ease of purification, the more active *E. coli*-produced rBoNT/A1 was either produced without additional tags or their tags were enzymatically removed after purification [49,50]. In addition, the linker region between the LC and HC was modified to include a thrombin cleavage site for easy conversion to the more active dichain [49]. These data, together with the results from our study, suggest that addition of protein tags, as well as proteolytic processing sites to rBoNT/As, may affect the resulting toxin’s potency. Furthermore, when proteins are purified, they undergo multiple chemical manipulations, some of which could affect toxin potency. Thus, effects from the purification procedures after heterologous protein production cannot be excluded as a factor in defining toxin potency. Finally, as-yet unknown post-translational modifications in *C. botulinum* also cannot be excluded as contributing factors. For heterologously produced rBoNTs, comparison to a wt rBoNT produced using the exact same method is thus essential in determining the effect of the specific amino acid alteration on toxicity.

Finally, various assays are used to determine toxicity of mutated rBoNTs. In this study, we used the in vivo mouse bioassay and a highly sensitive neuronal cell-based assay as two biologic assays, and we conducted assays in parallel with the native BoNT/A1. The greater residual toxicities of the individual mutant rBoNTs examined in this study (Table 1) compared to the previously indicated reductions in a specific functional property of the toxin using in vitro assays indicate an assay-specific discrepancy in determining residual toxicity. As with the in vivo mouse assays, the rat spinal cord-cell assays demonstrated that all the endogenous *Clostridium*-produced mutants in this study still had significant residual toxicity and retained the capacity to enter neuronal cells and cleave SNAP-25 proteins. Both in vivo mouse and in vitro cell assay results showed that the K759 mutation in the translocation domain of BoNT/A1 had no impact on residual toxicity (Figure 3), in spite of previous findings that the equivalent residue in tetanus toxin is essential for translocation [42]. Point mutations introduced to the ganglioside-binding and protein receptor-binding domain had a lower-than-expected impact: a ~110-fold reduction in EC50 in the rat spinal cord cells assay for both rBoNT/A1^W1266A^ and rBoNT/A1^G1292R^, and ~90- and ~25-fold decrease, respectively, by mouse bioassay (Table 1). In contrast, mutations targeting the light chain resulted in a greater decrease in toxin potency. Two mutations in the catalytic cleft, BoNT/A1^L175A,D370A^, resulted in an ~2000-fold decrease in SNAP-25 cleavage in primary rat spinal cord cells versus wild-type BoNT/A1 and a ~1800-fold reduction in LD50 in the mouse bioassays (Table 1). The greatest decrease in potency resulted from the two sets of mutations designed to prevent zinc coordination in the LC, rBoNT/A1^E224Q,R363A,Y366F^ and ciBoNT/A1, both of which had an ~12,500-fold decreased potency by mouse bioassay. In addition, both mutants were able to enter primary rat spinal cord cells and cleave SNAP-25 but did not reach 100% cleavage, indicating a severe defect in enzymatic activity (Figure 5). This same defect was also observed for 7M rBoNT/A1, which combines the L175A, D370A, E224Q, R363A, Y366F, K759A, and W1266A mutations (Figure 1). However, for our studies, until 100% cleavage is achieved, a comparative EC50 versus wt BoNT/A1 could not be determined for these mutant rBoNTs. The design of 7M rBoNT/A1 as a potential vaccine candidate was based on the hypothesis that iterative mutations in multiple functional domains of BoNT/A1 would have a multiplicative effect on reduction in toxin potency. While the 7M rBoNT/A1 had the lowest residual toxicity of all mutants tested, with an ~70,000-fold reduction in intraperitoneal mouse LD50 compared to wt BoNT/A1, the decrease in residual toxicity was not multiplicative for the individual sets of mutations (Figure 1). These data indicate that inactivating multiple functional sites in a biologically potent toxin like BoNT results in an additive but not multiplicative decrease in in vivo toxicity.

Our data suggest that factors other than introduced specific amino acid alterations may affect the residual toxicity of rBoNTs designed to be atoxic. While molecular pathways affecting the toxicity of heterologously produced rBoNTs are currently unknown, possible factors include the addition of protein tags for purification, production host-specific post-translational modifications, purification methods, and other amino acid modifications. Efforts to produce atoxic rBoNTs also have been extended to additional serotypes, predominantly using heterologous expression systems. Similar to heterologously produced mutated rBoNT/A constructs, the low residual toxicity (µg to mg per mouse range) observed for ciBoNT/B-F [32,53,54]; BoNT/C1 ad [55,56]; and chimeric BoNT/XA, XC, and XD [57] has primarily been attributed to the specific amino acid alterations introduced to the light-chain catalytic domain. In an effort to design a botulism vaccine, a 2017 study by Webb and associates produced ciBoNT/A-F in *Pichia pastoris*, which resulted in 100% mouse survival at 5–50 µg per mouse, depending on serotype [29,54]. Considering that the same ciBoNT/A produced in *C. botulinum* had residual toxicity at 50 ng/mouse (Figure 5), factors other than the H233A, E224A, and H227A mutations contribute to the lower residual toxicity of the same constructs produced in *Pichia*, which likely also applies to the other serotypes. Both detoxified and altered rBoNT/C1 ad produced in Sf9 cells, as well as hybrids of light chain-translocation domain of BoNT/X with heavy chain receptor-binding domain of either BoNT/A, BoNT/C, or BoNT/D, produced in *E. coli* were developed as neuronal therapeutic delivery systems and reported no mouse deaths at doses below 1 mg/mouse [55,56,57]. Given the toxicity data of the rBoNT/A1s generated in *C. botulinum* (Table 1) and the uncertainty surrounding the molecular pathways influencing the toxicity of rBoNTs produced in heterologous systems, it is imperative to exercise utmost caution when developing any catalytically inactive serotype for clinical application. Considering the lower sensitivity of some in vitro methods and the inability of the cell-based assay to determine relative EC50s for all mutants tested, here (Figure 2 and Figure 5), we recommend in vivo toxicity testing of new engineered mutated rBoNTs.

In summary, our study shows that the expression host and production method of mutated rBoNT/A1 may have a significant impact on toxicity of the resulting protein toxin. However, the work by Dolly and associates also shows that it is possible to produce a rBoNT/A1 in *E. coli* with similar potency as native BoNT/A1 [49,50]. This is an important consideration for toxicity evaluations of genetically inactivated protein toxin as vaccines, assay controls, or study materials. Our data suggest that the potency of new or mutated rBoNTs is affected by their production method, and that toxicity analyses should employ a comparative biologic toxin assay using wt BoNT produced under the exact same conditions.

## 4. Material and Methods

### 4.1. Biosafety, Biosecurity, and Ethics

Members of the Pellett laboratory are registered with the CDC Select Agent Program for research involving botulinum neurotoxins and botulinum neurotoxin-producing strains of *Clostridia*. The Pellett laboratory research program; its procedures, occupational health plan, documentation, and security; and all related facilities are meticulously monitored by the University of Wisconsin–Madison Biosecurity Task Force, University of Wisconsin–Madison Office of Biological Safety, and the University of Wisconsin Select Agent Program. As a member of the University of Wisconsin–Madison Select Agent Program, the Pellett laboratory is inspected at regular intervals by the CDC and the Animal and Plant Health Inspection Services. All personnel have undergone suitability assessments and completed thorough and continuing biosafety training. All personnel are trained in biosafety level 3 (BSL3) and select agent practices before participation in any laboratory study that involves botulinum neurotoxins and neurotoxigenic *C. botulinum* strains. All recombinant DNA-related protocols for the synthesis of the recombinant, mutant BoNT genes, and their subsequent expression in *C. botulinum* strains were approved by the University of Wisconsin Institutional Biosafety Committee (IBC), with specific experiments approved by the Division of Select Agents and Toxins at the CDC (protocol # B00000934, approved 18 March 2024). A dual-use research of concern (DURC) risk mitigation plan was prepared and implemented by the University of Wisconsin Madison Select Agent Program, and NIAID for these experiments. Preparation of all the recombinant BoNT gene constructs was performed at biosafety level 2, while experiments involving transfer of gene expression vectors into the *C. botulinum* expression host strain and purification of the recombinant BoNT were performed in a biosafety level 3 (BSL3) facility, as described in the CDC/NIH documents and in accordance with all Select Agent regulations. Animal experiments involving BoNT select agents were approved by the University of Wisconsin–Madison Institutional Animal Care and Use Committee (IACUC; protocol number M006326-R01-A01; approval date, 14 April 2023).

### 4.2. Reagents

Oligonucleotide primers were purchased from IDT (Corralville, IA, USA; Table 2). PCR reactions used either the Phusion High-Fidelity Master mix with HF buffer or Phusion Hot Start Flex 2X Master mix (both from New England Biolabs, Ipswich, MA, USA). Cloning-related enzymes (Restriction endonucleases), Quick CIP (calf alkaline phosphatase), DNA ligase, and chemically competent *Escherichia coli* DH10β cells were all obtained from New England Biolabs (Ipswich, MA, USA). Mutations were introduced into the wild-type BoNT/A1 DNA sequence using the QuikChange Lightning Multi site-directed mutagenesis kit (Agilent Technologies, Santa Clara, CA, USA). Antibiotics used in these experiments (carbenicillin, chloramphenicol, thiamphenicol, erythromycin, and cycloserine) were purchased from Sigma-Aldrich (St. Louis, MO, USA).

### 4.3. Bacterial Strains and Media

*E. coli* strains were incubated at 37 °C in either Luria Broth (LB) media or plates supplemented with the required antibiotics for plasmid selection. *E. coli* strain CA434 (kindly provided by Dr. N. Minton, University of Nottingham, UK) functioned as the donor strain during conjugal expression vector transfer experiments from *E. coli* into *C. botulinum*. Antibiotics were used in the following final concentrations in *E. coli* strains: carbenicillin at 50 µg/mL, kanamycin at 50 µg/mL, and chloramphenicol (at either 25 µg/mL in agar plates or 12.5 µg/mL in liquid media).

*C. botulinum* strains were maintained in 5% Trypticase peptone, 0.5% Bacto peptone, 0.4% glucose, 2% yeast extract, and 0.1% L-cysteine HCl, pH 7.3–7.4 (TPGY) liquid or agar media. For production of toxin, the clostridial strains were grown in Toxin Production Media (TPM; 2% Phyto peptone, 1% yeast extract, 0.5% glucose, pH 7.3). All *clostridium* strains were incubated anaerobically in an anaerobic chamber Bactron 600 (Sheldon Manufacturing, Cornelius, OR, USA), with an atmosphere of 90% N_2_, 5% CO_2_, and 5% H_2_. The cultures were grown statically in bottles or nitrogen-flushed Hungate tubes. Antibiotics were used at the following concentrations: cycloserine at 250 µg/mL and thiamphenicol at 15 µg/mL.

### 4.4. Botulinum Neurotoxins

The wild-type 150-kDa BoNT/A1 toxin, free of any complexing proteins, was isolated from wild-type *C. botulinum* strain Hall A-hyper, as previously described [45]. All recombinant, mutant 150 kDa BoNT/A1 toxins, free of any complexing proteins, were produced and purified from an endogenous *C. botulinum* expression system using a similar purification method [44].

For production of mutated BoNT/A1, the wild-type BoNT/A1 gene was amplified via PCR using total genomic DNA isolated from *C. botulinum* strain Hall A-hyper (GenBank accession number AF461540), and Phusion Hot Start Flex 2X Master mix per manufacturer’s instructions (New England Biolabs, Ipswich, MA, USA). For each mutant BoNT/A1 gene, the indicated mutation(s) were introduced using specific primers (Table 2) and the Agilent QuikChange Lightning Multi site-directed mutagenesis kit (Agilent Technologies, Santa Clara, CA, USA) according to manufacturer’s instructions. The presence of nucleotide substitutions in the recombinant genes was confirmed by Sanger sequencing at the University of Wisconsin Biotechnology Center or by whole plasmid sequencing with a Minion Nanopore Sequencing system (Oxford Technologies, Abingdon, UK) per manufacturer’s instructions. In brief, samples were barcoded per manufacturer’s instructions, pooled, quantified via Qbit (Thermofisher, Waltham, MA, USA), purified with magnetic AMPure XP Beads (Sigma-Alrich, St. Louis, MO, USA), and loaded onto the Nanopore flongle flow cell (all per manufacturer’s instructions). The generated sequences were then analyzed on either SnapGene Version 6.2 (GSL Biotech LLC, Boston, MA, USA) or MacVector Version 18.6 (MacVector Inc., Apex, NC, USA), as well as UGENE bioinformatics software version 46 (Unipro, Novosibirsk, Russia) [58].

All recombinant mutant BoNT/A1 genes were then inserted into the clostridial expression vector pMTL83152 and transformed into *E. coli* strain CA434. The resulting *E. coli* transformants were then mated with the nontoxigenic *C. botulinum* expression strain Hall A-hyper/tox^−^ via conjugation, as previously described [44]. The recombinant clostridial expression vectors with mutant toxin genes were isolated from *C. botulinum* expression strain Hall A-hyper/tox^−^, and the sequence of the mutated genes was confirmed again by sequencing, as described above.

The recombinant mutant proteins were produced in the *C. botulinum* expression strain Hall A-hyper/tox^−^ in TPM supplemented with 15 µg/mL thiamphenicol at 37 °C for 4 days. The recombinant toxin was then purified as previously described [44,45], and purified 150 kDa toxins were stored in phosphate-buffered saline with 40% glycerol at −20 °C. Toxin concentrations were determined via absorbance measurements at A278 with an extinction coefficient of 1.63 for 1 mg/mL in a 1 cm light path [59].

### 4.5. Primary Rat Spinal Cord (RSC) Cells

Preparation of primary rat spinal cord (RSC) cells for cell assays was performed as previously described [60,61,62]. In brief, a female rat, Sprague Dawley (Harlan Sprague-Dawley, Indianapolis, IN, USA), at the E15 gestational stage was euthanized via CO_2_, and the pups from the uterus were promptly placed in a dish containing dissection medium (Hanks balanced salt solution, with 10 mM HEPES and 20 mM Glucose). While in this solution, the pups were immediately decapitated, and their spinal cords were isolated. The membranes and ganglia surrounding the spinal cords were then gently removed and transferred to 4.5 mL of fresh dissection media. The spinal cords were minced with sterile scissors and tweezers, and cells were dissociated by addition of 1 mL of TrypLE (Invitrogen, Waltham, MA, USA) and incubated for 10 min at 37 °C. The trypsinization solution was removed, and rat spinal cords were washed once with 15 mL of dissection media. After the minced, trypsinized spinal cords settled out of solution, the washing dissection medium was removed, and 1 mL of culture medium (Neurobasal medium with Glutamax, B27, and penicillin/streptomycin 50 U/mL (final concentration); all from Gibco, Waltham, MA USA) pre-warmed to 37 °C was added, and cells were triturated via pipetting up and down 8–10 times. Live cells were quantified via trypan blue exclusion assay with a Countess Automated Cell Counter (Invitrogen, Waltham, MA, USA) and were plated onto a 96-well tissue-culture microplate (Mfg: TPP, Trasadingen, Switzerland) coated with 0.01% poly-L ornithine (Sigma-Aldrich, St. Louis, MO, USA) and Matrigel matrix (Corning, Glendale, AZ, USA) at ~75,000 cells per well. Cells were then incubated at 37 °C with 5% CO_2_ for at least 17 days prior to use to allow for differentiation, with biweekly media changes.

### 4.6. Cell-Based BoNT Assay

The indicated quantities of purified BoNT/A1 toxin were added to a total volume of 50 µL culture media per well and incubated for 48 h at 37 °C with 5% CO_2_. After incubation, cells were lysed with 75 µL of 1X LDS lysis buffer (Invitrogen, Waltham, MA, USA). Sample and control (no toxin added) cells were analyzed by SDS-PAGE electrophoresis with 12% NuPAGE Novex Bis-Tris gels in 1X NuPAGE MES running Buffer (all from Invitrogen). Proteins were then transferred onto a 0.45 µm PVDF membrane via a semi-dry transfer (Invitrogen Novex Thermo Horizontal Semi-Dry Blotter). Full-length and cleaved SNAP-25 was detected with mouse anti-SNAP-25 monoclonal antibodies that recognizes both uncleaved and cleaved SNAP-25 equally (Synaptic Systems, Göttingen, Germany), and a secondary goat, anti-mouse, AP-conjugated antibody (Seracare, Milford, MA, USA). After development via Phosphaglo^TM^ AP substrate (SeraCare, Milford, MA, USA), all blots were imaged on an Azure Biosystems C600 imager. The percent of cleaved versus uncleaved SNAP-25 was quantified via densitometry with Azurespot (Version 2.0.062) within the linear dynamic range. The percent cleaved SNAP-25 vs. dose was used to create a nonlinear four-parameter regression curve fit and determine the EC50 (Half Maximal Effective Concentration) in GraphPad Prism (Version 9; *n* ≥ 3). Graphs showing the average and standard deviation of the data points, as well as the regression curve, were generated in GraphPad Prism (Version 9).

### 4.7. Specific Toxin Activity Determination

The specific activity of all BoNTs (recombinant and wild type) was ascertained by (in vivo) mouse bioassay (MBA) [63,64]. Serial dilution of either the wild-type or mutant BoNTs in 30 mM sodium phosphate buffer pH 6.3 with 0.2% gelatin (Gelphos buffer) were administered by intraperitoneal injection into 6 groups of 4 female mice (0.5 mL/mouse). Mice were observed for up to 6 days, and each toxin’s specific activity was determined via the methods by Reed and Muench [65]. The specific activity for all toxins is expressed as picogram (pg) of toxin/LD50 (LD50 = median lethal dose).

## Figures and Tables

**Figure 1 ijms-25-08955-f001:**
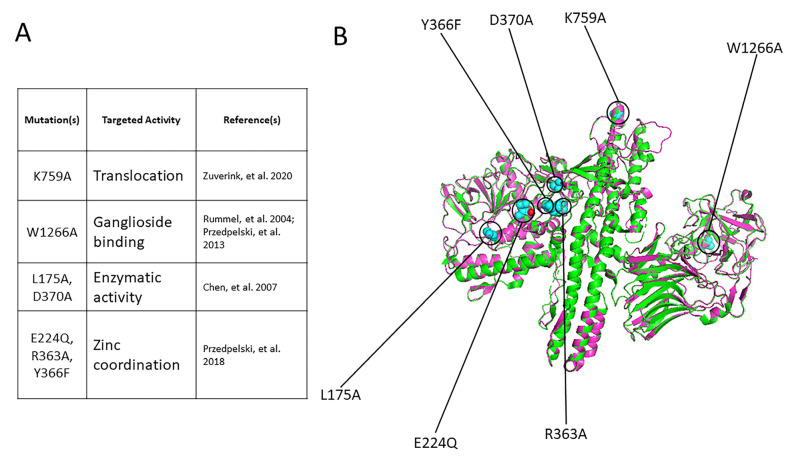
Design of 7M BoNT/A1. The mutations in 7M rBoNT/A1 are based on several different published mutations in clostridial neurotoxins combined into a single BoNT/A1 holotoxin. A single-point mutation in the ganglioside binding domain (W1266A), as well as translocation domain (K759A), was introduced into the ~100 kDa heavy chain [39,42]. Two sets of mutations were placed in the metalloprotease light chain: a double mutation (L175A and D370A) in the enzymatic cleft targeting hydrolytic activity and three-point mutations (E224Q, R363A, and Y366F) introduced into the zinc coordination motif, which has shown to significantly abrogate toxin activity and potency in an *E. coli*-produced rBoNT/A1^E224Q,R363A,Y366F^ [30,40]. (**A**) Specific mutation(s) that make up the 7M BoNT/A1 design, including their target and published origin. (**B**) Overlay of 7M BoNT/A1 (magenta) vs. wt BoNT/A1 (green). The altered amino acid residues in 7M rBoNT/A1 are shown in cyan spheres. The zinc ion of the wild-type BoNT/A1 LC is shown in red. The 7M BoNT/A1 amino acid sequence was inputted into Phyre 2 [43] to generate a structural model using intensive mode. This structural model of 7M BoNT/A1 crafted in Phyre2 [43] was subsequently superimposed with the wild-type BoNT/A1 (PDB ID 2NYY) using PyMol [41]. PyMOL Molecular Graphics System, Version 4.6 Schrödinger, LLC (New York, NY, USA).

**Figure 2 ijms-25-08955-f002:**
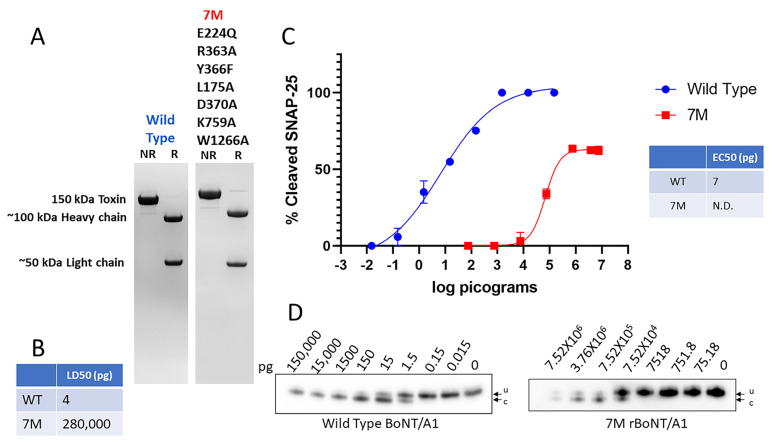
The purity and biological activity of *Clostridium* produced wild-type and recombinant 7M rBoNT/A1. Wild-type BoNT/A1 was produced in *C. botulinum* in strain Hall A-hyper, while the mutant 7M rBoNT/A1 was produced in *C. botulinum* strain Hall A-hyper/tox^−^. Both were purified using biochemical methods. 7M rBoNT/A1 was successfully produced in an endogenous clostridial expression system, which also produces all BoNT/A complexing proteins but no wild-type toxin. The 7M rBoNT/A1 was also successfully synthesized and purified without the use of purification tags or additional modifications [44]. (**A**) Coomassie-stained SDS-PAGE gels of reduced (R) and non-reduced (NR) samples of purified wild-type and 7M rBoNT/A1. (**B**) Specific toxicity of wild-type BoNT/A1 and 7M rBoNT/A1 in mice. (**C**) Graphic representation of Western blot data analyzing potency of wt BoNT/A1 and 7M rBoNT/A1 in primary rat spinal cord cells. The cultured neurons were exposed to serial dilutions of the indicated toxins for 48 h, and the percent SNAP25 cleavage was determined in cell lysates by Western blot, using an anti-SNAP-25 antibody that equally detects both uncleaved and cleaved SNAP-25, and densitometry. Values were then plotted in GraphPad Prism Version 9 (*n* ≥ 3), and the resulting graphs and EC50 (Half Maximal Effective Concentration) values were derived from a nonlinear four-parameter regression curve fit. A comparative EC50 for 7M rBoNT/A1 could not be determined (N.D.) since 100% cleavage was not achieved under experimental conditions. (**D**) Representative Western blots of the primary RSC cell-based Assays. u: uncleaved SNAP-25, c: cleaved SNAP-25. All Western blots used to determine the EC50 are shown in Appendix A.

**Figure 3 ijms-25-08955-f003:**
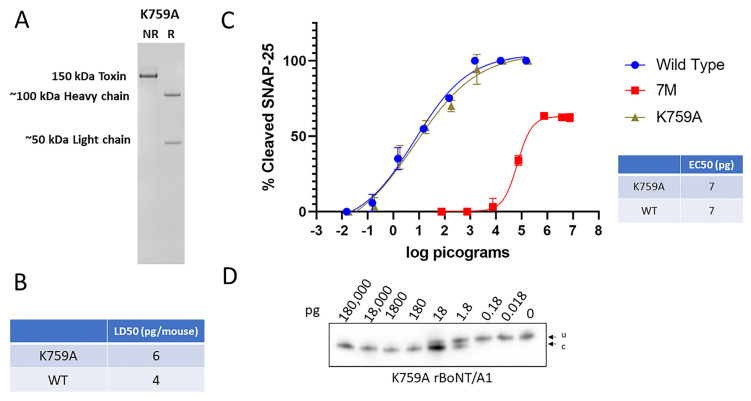
The purity and biological activity of *Clostridium* produced rBoNT/A1^K759A^. The mutant BoNT/A1^K759A^ was produced in *C. botulinum* strain Hall A-hyper/tox^−^ and purified using biochemical methods. (**A**) Coomassie-stained SDS-PAGE gels of reduced (R) and non-reduced (NR) purified samples of rBoNT/A1^K759A^. (**B**) Specific toxicity of this rBoNT/A1^K759A^ in mice, compared to wt BoNT/A1. (**C**) Graphic representation of Western blot data analyzing potency of rBoNT/A1^K759A^ relative to wt BoNT/A1 and 7M BoNT/A1 in primary rat spinal cord cells. The cultured neurons were exposed to serial dilutions of the indicated toxins for 48 h, and the percent SNAP25 cleavage was determined in cell lysates by Western blot, using an anti-SNAP-25 antibody that equally detects both uncleaved and cleaved SNAP-25, and densitometry. Values were then plotted in GraphPad Prism Version 9 (*n* ≥ 3), and the resulting graphs and EC50 (Half Maximal Effective Concentration) values were derived from a nonlinear four-parameter regression curve fit. (**D**) Representative Western blots of the primary RSC cell-based Assays. u, uncleaved SNAP-25; c, cleaved SNAP-25. All Western blots used to determine the EC50 are shown in the Appendix A.

**Figure 4 ijms-25-08955-f004:**
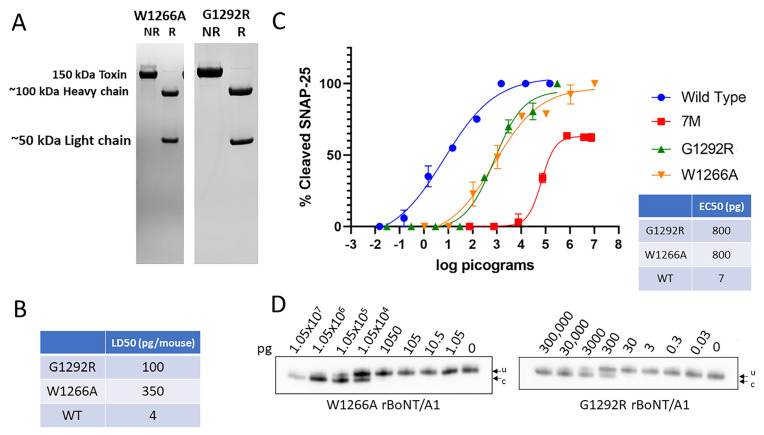
The purity and biological activity of *Clostridium*-produced rBoNT/A1 with mutations targeting the binding capabilities of the HC. All mutant rBoNT/A1s were produced in *C. botulinum* strain Hall A-hyper/tox^−^ and purified using biochemical methods. (**A**) Coomassie-stained SDS-PAGE gels of reduced (R) and non-reduced (NR) samples of purified rBoNT/A1^W1266A^ and rBoNT/A1^G1292R^. (**B**) Specific toxicity of both mutant holotoxins in mice, compared to wt BoNT/A1. (**C**) Graphic representation of Western blot data analyzing potency of rBoNT/A1^W1266A^ and rBoNT/A1^G1292R^ relative to wt BoNT/A1 and 7M BoNT/A1 in primary rat spinal cord cells. The cultured neurons were exposed to serial dilutions of the indicated toxins for 48 h, and the percent SNAP25 cleavage was determined in cell lysates by Western blot, using an anti-SNAP-25 antibody equally detects both uncleaved and cleaved SNAP-25, and densitometry. Values were then plotted in GraphPad Prism Version 9 (*n* ≥ 3), and the resulting graphs and EC50 (Half Maximal Effective Concentration) values were derived from a nonlinear four-parameter regression curve fit. (**D**) Representative Western blots of the primary RSC cell-based Assays. u, uncleaved SNAP-25; c, cleaved SNAP-25. All Western blots used to determine the EC50 are shown in Appendix A.

**Figure 5 ijms-25-08955-f005:**
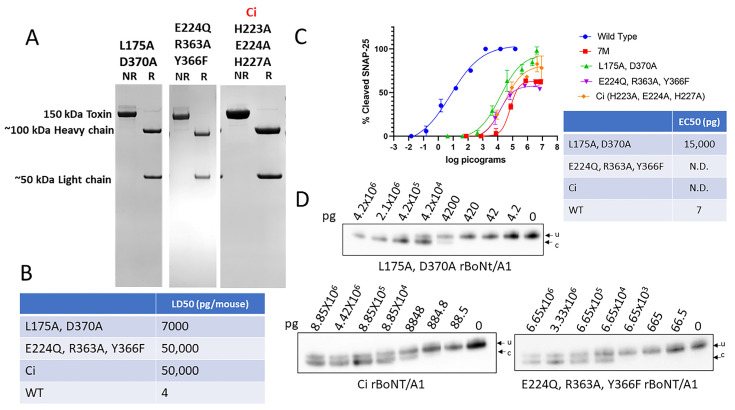
The purity and biological activity of all *Clostridium*-produced rBoNT/A1 with mutations targeting the different activities of the BoNT/A1 LC. All mutant rBoNT/A1s were produced in *C. botulinum* strain Hall A-hyper/tox^−^ and purified using biochemical methods. (**A**) Coomassie-stained SDS-PAGE gels of reduced (R) and non-reduced (NR) samples of purified rBoNT/A1^L175A,D370A^ that abrogate the LC enzymatic activity, as well as rBoNT/A1^E224Q,R363A,Y366F^ and ciBoNT/A1, which abolish the zinc ion coordination of the LC metalloprotease. (**B**) Specific toxicity of all mutant holotoxins in mice, compared to wt BoNT/A1. (**C**) Graphic representation of Western blot data analyzing potency of rBoNT/A1^L175A,D370A^, rBoNT/A1^E224Q,R363A,Y366F^, and rBoNT/A1^H223A,E224Q,H227A^ (ciBoNT/A1) relative to wt BoNT/A1 and 7M BoNT/A1 in primary rat spinal cord cells. The cultured neurons were exposed to serial dilutions of the indicated toxins for 48 h, and the percent SNAP25 cleavage was determined in cell lysates by Western blot, using an anti-SNAP-25 antibody that equally detects both uncleaved and cleaved SNAP-25, and densitometry. Values were then plotted in GraphPad Prism Version 9 (*n* ≥ 3), and the resulting graphs and EC50 (Half Maximal Effective Concentration) values were derived from a nonlinear four-parameter regression curve fit. A comparative EC50 for rBoNT/A1^E224Q,R363A,Y366F^ and rBoNT/A1^H223A,E224Q,H227A^ (ciBoNT/A1) could not be determined (N.D.) since 100% cleavage was not achieved under experimental conditions. (**D**) Representative Western blots of the primary RSC cell-based assays. u, uncleaved SNAP-25; c, cleaved SNAP-25. All Western blots used to determine the EC50 are shown in Appendix A.

**Table 1 ijms-25-08955-t001:** Comparison of Mouse LD50 (pg/mouse) of endogenously produced mutant BoNT/A1 vs. the heterologously produced counterpart.

	Mouse LD50 [pg/mouse]	
BoNT/A1 Genotype	Produced in *C. botulinum*	Produced in Heterologous Host/Previously Reported Data
Wild type	4	~4 [49,50]
K759A	6	n/a
W1266A	350	800 * [38,39]
L175A, D370A	7000	n/a
E224Q, R363A, Y366F	50,000	>10,000,000 [30]
E224Q, R363A, Y366F, L175A, D370A, K759A, W1266A, (7M)	280,000	n/a
H233A, E224A, H227A (Ci)	50,000	>50,000,000 [29]
G1292R	100	1200 * [46]

* Estimated based on mouse phrenic nerve-assay data [46]. Also, W1266 was mutated to L in the initial study [38].

**Table 2 ijms-25-08955-t002:** Primers to introduce point mutations into BoNT/A1 gene.

Oligonucleotide Name	Sequence (5′-3′)	Utility
A1LC-F(Nde)	GCCATATGCCATTTGTTAATAAACAATTTAATTATAAAGATCC	Amplification of the entire BoNT/A1 gene from genomic DNA isolated from *C. botulinum* strain Hall A-hyper
A1HC-R(Nhe)	GCGCTAGCTTACAGTGGCCTTTCTCCCCATCCATCATCTAC	
A1M_505-542	GGACATGAAGTTTTGAATGCTACGCGAAATGGTTATGG	Mutagenesis primer for substitution of L175A
A1M_652-686	GCAGTAACATTAGCACATCAACTTATACATGCTGG	Mutagenesis primer for substitution of E224Q
A1M_1063-1125	GTTAAGTTTTTTAAAGTACTTAACGCAAAAACATTTTTGAATTTTGCTAAAGCCGTATTTAAG	Mutagenesis primer for substitutions of R363A; Y366F; D370A
A1M_2257-2302	CAA TAT ACT GAG GAA GAG GCA AAT AAT ATT AAT TTT AAT ATTGATG	Mutagenesis primer for substitution of K759A
A1M_3781-3821	CTA GTA GCA AGT AAT GCT TAT AAT AGA CAA ATA GAA AGA TC	Mutagenesis primer for substitution of W1266A

Underlined: restriction sites for either *Nde*I (CATATG) or *Nhe*I (GCTAGC). Red denotes substituted nucleotide sequences to introduce indicated point mutations.

## Data Availability

Data are contained within the article. Additional data are available subject to Select Agent Restrictions.

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
