# Peer review of "Potency Evaluations of Recombinant Botulinum Neurotoxin A1 Mutants Designed to Reduce Toxicity"

_ijms, 2024, doi:10.3390/ijms25168955_

Round 1
Reviewer 1 Report
Comments and Suggestions for Authors
It is a well-designed and excellently written paper, should be published immediately after a few minor Typo correction: 1). Line 127 on page 3, YR63A should be R363A; 2). Line 299 on page 9, ref [47] should be [44]; 3). Line 677 and 685 on page 17 for ref [44] and [47] is the same paper, delete one; 4). Rearrange all the reference after delete [47]
As a dual use unique agent, natural or engineered BoNT has the potential to be used as powerful curing drugs for many diseases, or to be misused as bioterrorist weapons to cause mass casualty.
To address the issues related with extremely high potency toxicity of BoNT while minimizing the impact on the immunogenicity in the process of vaccine search, many researchers and scientists have tried to eliminate the normal biological function of BoNT via structure-guided molecular engineering. And some mutations on the LC or HC domains of BoNT were found to be satisfactory in the direction of rendering BoNT molecular atoxic.
But whether multiple mutations on different toxin domains combined will have synergetic effect has never been tested systematically in parallel using authentic in vivo and in vitro evaluation methods.
This paper addressed the toxicity evaluation issues for seven mutants of BoNT/A1, and by comparing the same mutant produced heterogeneously or endogenously, clear differences were noticed and the possible reasons discussed for the 7 mutants.
Sufficient references were provided in the paper , and the data is well presented and solid, conclusions are reasonable, and the information will help guide further development of BoNT application in multiple fields. So this paper should be published immediately after minor revisions as listed.
Comments on the Quality of English Language
Excellently written paper, please make the minor correction as indicated above
Author Response
We thank the reviewer for their positive assessment of our manuscript and for the helpful comments. We have made all requested changes as outlined in the attached file.

Reviewer 2 Report
Comments and Suggestions for Authors
In “Toxicity Evaluations of Botulinum Neurotoxin A1 Mutants Designed to be Atoxic”, Viravathana et al conduct cell-based and in vivo murine lethality studies to assess the effects of various mutations of BoNT/A on toxicity. They focus on a seven-mutant variant containing one mutation in the translocation domain (K759A), one in the ganglioside binding domain (W1266A) and five mutations in the LC (E224Q, R363A, Y366F, L175A, D370A) and assess the impact of various sets of mutations on toxicity. The manuscript is well written, and the data provided is clearly explained. The combination of in vitro and in vivo toxicity studies is compelling. However, this manuscript is largely descriptive and a bit unfocused. The manuscript would strongly benefit from additional studies to understand the main finding that all mutants expressed in C. botulinum had 100-1000x greater toxicity than similar (but not identical) mutants expressed in E coli (solely based on reports by other laboratories). The authors provide several hypotheses to explain the putatively greater toxicity of mutants produced in C. botulinum vs E. coli, including (1) interference by protein tags and/or proteolytic substrates, (2) structural changes due to the use of affinity-based purification steps, and/or (3) hypothetical differences in post-translational processing. A fourth possibility that the authors must address is contamination of their recombinant bacterial stocks or purified protein by small quantities of wild-type C. botulinum or active toxin. This is certainly possible since the authors run a select agent facility focused on expression of wild-type BoNTs and purification of wild-type and recombinant toxins using tag-free methods, which will not purify recombinant toxin away from wild-type toxin.
To be suitable for publication, the authors should directly test their hypotheses. They do not need to be conducted using the 7M variant given its poor expression in E. coli, but could be conducted using tagged and untagged E224Q, R363A, Y366F variants expressed in C. botulinum and E. coli and purified both using affinity-based vs biochemical methods for direct comparisons. The catalytic activity of the various constructs and purification methods should also be directly compared in vitro using the BoTest assay or Endopep-MS. Without these additional studies, the manuscript is primarily descriptive in nature and does not present a complete story.
Minor points:
1. It should be noted that Vazquez-Cintron et al 2017 (Scientific Reports), McNutt et al 2021 and Miyashita et al 2021 (both translational medicine) demonstrated recombinant expression of ciBoNTs from insect cells and E. coli, with affinity purification tags and multiple purification steps. Vazquez-Cintron showed a complete lack of proteolytic activity for a triple mutant ciBoNT/C using a MS assay in collaboration with the CDC. McNutt et al (Figure 1) showed a complete lack of SNAP-25 cleavage in a BoTest assay even at very high concentrations of ciBoNT/C, despite immunoblot evidence of ciBoNT/C interactions with SNAP-25. Miyashita showed a complete lack of catalytic activity in primary neurons. They also demonstrated intraneuronal therapeutic activity against wild-type BoNT/A1. Although these studies used serotypes other than BoNT/A, they argue that the use of affinity tags, proteolytic substrates and complex purification strategies does not necessarily interfere with toxin uptake and internalization.
2. Line 60: Given the widespread use of BoNT serotypes A, B and soon E for therapeutic indications, I’m not sure there is a crucial need for new vaccines for the ‘at-risk public’. I suggest this statement be toned down and/or a specific use case for vaccines be made.
3. Line 127: YR63A should be R363A
4. Demonstrating the lack of effect of the K759A mutation was interesting.
5. Line 361: Suggest ‘additive’ rather than ‘interative’.
6. Line 383: suggest using ‘may have’ rather than ‘can have’ unless the above hypotheses are tested
Author Response
We thank the reviewer for their assessment and suggestions to improve our manuscript. We are addressing each individual suggestion in the attached file.

Reviewer 3 Report
Comments and Suggestions for Authors
In this manuscript entitled "Toxicity Evaluation of Botulinum Neurotoxin A1 Mutants Designed to be Atoxic", the authors propose that the production systems significantly impact the residual toxicity of inactivated recombinant BoNTs (rBoNTs). The authors' significant finding in this study is that the toxicity of recombinant BoNTs produced by the Clostridium botulinum expression system is much higher than the E. coli system. While the reviewer believes that the authors evaluated important issues for toxicity of rBoNTs, their data does not fully support the author's conclusion. There are major flaws that prevent publication of manuscript at this stage.
Major comments
1. The authors' main conclusion is that the expression system of rBoNTs impacts their toxicity. However, the most crucial experiment missing in this study is a direct comparison to see if there are significant differences in the toxicity of rBoNTs purified from E. coli (or yeast) and C. botulinum systems. This direct comparison is crucial as it could provide critical insights into the role of the production system in the toxicity of rBoNTs. The authors should purify several rBoNT/A1 from the E. coli expression system and compare the toxicity of the same mutants purified from their C. botulinum expression system. Since at least one of the mutants (Ci; E224Q, R363A, Y366F) can be expressed in E. coli (Przedpelski, 2018), it would be possible to compare these in cell-based assay and mouse bioassays. This direct comparison could enhance the robustness of the authors' findings.
2. Results (western blot): Most image data from Western blots are of low quality because the signal intensity is weak and blurry. The lack of loading controls in all figures makes it difficult to evaluate the results. To improve the data quality, the authors should consider including loading controls in their figures and provide another blot with a longer exposure time, enhancing the clarity and reliability of their results.
3. Results (Fig. 2D, 7M rBoNT/A1): The band intensity of 7.52X106 and 3.76X106 does not seem 60% cleavage of SNAP-25 in Fig. 2C. It is not clear from Methods how the quantification of cleaved SNAP-25 was carried out (line 515). Raw data should accompany the quantification of densitometry of cleaved SNAP-25.
4. The reviewer is concerned that the exact data for the control, wild-type BoNT/A, is used in mouse bioassays and cell-based assays across Figures 2-5. The authors should clearly state how the experiments were performed and why the same controls are diverted to different figures.
5. Line 148: The details of how to express 7M rBoNT/A1 in E. coli are lacking in the Method section. It would be helpful if the authors would explain the details of the production of it.
6. The abbreviation FSAP in Table 1 is not defined.
Minor comments
1. Correct Table 2 to 1 (Line 353).
2. Correct Table 1 to 2 (Line 459).
Author Response
We would like to thank the reviewer for careful analysis of our manuscript and insightful comments. Individual responses are shown in the attached file

Reviewer 4 Report
Comments and Suggestions for Authors
The authors have written the paper well, however, I have some questions-
1. How do you explain the precise molecular pathways that cause the greater residual toxicity found with the endogenously produced 7M rBoNT/A1 relative to previous E. coli productions?
The manuscript explores the unsuccessful endeavors to generate 7M rBoNT/A1 in E. coli as a result of challenges related to insolubility. Could you provide more details on the precise difficulties faced and any possible approaches to address these problems in future recombinant productions?
3. The incorporation of several mutations in 7M rBoNT/A1 appears to have resulted in an interaction effect that did not cause a proportional decrease in toxicity. Can additional computational or structural analysis provide more clarity on the relationships between these mutations?
4. The study used a C. botulinum expression system that naturally produces complexing proteins. How might these proteins influence the toxicity and efficacy of the mutated toxins, and were they considered in the analysis?
5. You reported a significant decrease in toxicity with mutations preventing zinc coordination in the light chain. However, enzymatic activity was not completely eliminated. What additional modifications could potentially lead to complete inactivation?
6. Given the variability in toxin potency based on the expression system, what steps can be taken to standardize production methods to ensure consistent quality and safety of vaccine candidates?7.The EC50 values for certain mutant toxins could not be obtained due to the inability to accomplish complete cleavage of SNAP-25. What alterations to the experimental design could enable a comprehensive evaluation of the complete spectrum of potency exhibited by these mutants?
8. Could you provide an analysis of the possible immune response that these rBoNT/A1 mutants may elicit, particularly in relation to their remaining enzymatic capabilities?
9. How do the variations in toxicity between proteins produced naturally and those produced in different hosts impact the applicability of the findings to other BoNT/A serotypes or neurotoxins?
10. Is there a strategy to investigate alternate methodologies that could utilize a reduced number of very potent mutations to achieve comparable or enhanced safety profiles while maintaining immunogenicity, given the intricate nature of mutations and their impacts?
Author Response
We thank the reviewer for their positive assessment and the raised questions. We have responded to the individual questions

Round 2
Reviewer 2 Report
Comments and Suggestions for Authors
1. I did not say the "conducting work in a Select Agent registered facility would somehow *increase* the likelihood of contamination". I said the fact that work was (a) conducted in a select agent facility where wild-type toxin is being produced and (b) involved a purification scheme that did not exclude recovery of wild-type toxin results in the potential for cross-contamination of recombinant toxin protein with small quantities of wild-type toxin protein, which would only be apparent at high doses. That is a simple explanation for the increased toxicity of C. botulinum-expressed protein vs E. coli-expressed protein. Simply saying the difference could nto be caused by cross-contamination is not a scientific explanation.
2. Without some testing of their hypotheses, much of the manuscript is of questionable value to the broader community.
Author Response
Please see our response to the two comments by the reviewer below:
- I did not say the "conducting work in a Select Agent registered facility would somehow *increase* the likelihood of contamination". I said the fact that work was (a) conducted in a select agent facility where wild-type toxin is being produced and (b) involved a purification scheme that did not exclude recovery of wild-type toxin results in the potential for cross-contamination of recombinant toxin protein with small quantities of wild-type toxin protein, which would only be apparent at high doses. That is a simple explanation for the increased toxicity of C. botulinum-expressed protein vs E. coli-expressed protein. Simply saying the difference could nto be caused by cross-contamination is not a scientific explanation.
We thank the reviewer for that clarification. However, we still respectfully disagree. We do not understand how using the clostridial expression system specifically would contribute to the risk of cross-contamination with wt toxin. An appropriate scientific approach with any expression system would include direct comparison with wt toxin produced in the same system, so the likelihood of cross-contamination should be the same for any system. With literally any comparative mutant protein analysis, the risk of cross-contamination exists. Furthermore, in our own lab the E.coli produced mutant toxins had much lower potency than Clostridium produced toxins, as cited in the manuscript, and the same risk for cross-contamination exists for both E.coli produced and Clostridium produced rBoNTs within the same lab. We are of course aware and very concerned with the risk of cross-contamination with this and any other study, and we are taking many precautions to avoid such risk: The mutant rBoNTs are purified separately from wt BoNT purification, all chromatography media used is always fresh, all buffers are prepared fresh, all work surfaces are disinfected and cleaned after each use, activities potentially producing aerosols are conducted in a BSC, and several rBoNTs have been prepared more than once with equal toxicity results.
Overall, we do not feel the manuscript will be improved or more accurate by including this common notion into the discussion, and we in fact feel it would be misleading.
2. Without some testing of their hypotheses, much of the manuscript is of questionable value to the broader community.
We strongly and respectfully disagree with this assessment (as did the other reviewers) and request the editor to make a decision on the value of the manuscript to the broader community.
Reviewer 4 Report
Comments and Suggestions for Authors
The authors have replied to my question satisfactorily.
Author Response
The authors have replied to my question satisfactorily.
We thank the reviewer for their original comments that helped improve our manuscript and the positive assessment of our response.